# Influence of Deposition Modes and Thermal Annealing on Residual Stresses in Magnetron-Sputtered YSZ Membranes

**DOI:** 10.3390/membranes12030346

**Published:** 2022-03-19

**Authors:** Andrey Solovyev, Sergey Rabotkin, Anna Shipilova, Dmitrii Agarkov, Ilya Burmistrov, Alexander Shmakov

**Affiliations:** 1Institute of High Current Electronics SB RAS, 634055 Tomsk, Russia; rabotkin@yandex.ru (S.R.); lassie2@yandex.ru (A.S.); 2Osipyan Institute of Solid State Physics RAS, Chernogolovka, 142432 Moscow, Russia; dima.agarkov@gmail.com (D.A.); i.n.burmistrov@gmail.com (I.B.); 3Budker Institute of Nuclear Physics SB RAS, 630090 Novosibirsk, Russia; highres@mail.ru

**Keywords:** solid oxide fuel cell, yttria-stabilized zirconia, electrolyte, residual stress, high temperature annealing

## Abstract

Thin-film electrolyte made of 8-mol% yttria stabilized zirconia (8YSZ) for solid oxide fuel cells (SOFCs) was fabricated on anode substrates using reactive magnetron sputtering of Zr-Y targets in a mixture of Ar and O_2_ gases. The deposition of 4–6 µm thin-film electrolyte was in the transition or oxide modes differing by the oxygen concentration in the sputtering atmosphere. The half-cell bending of the anode-supported SOFCs was measured to determine the residual stresses in the electrolyte films after the deposition and thermal annealing in air. The dependences were studied between the deposition modes, residual stresses in the films, and the SOFC performance. At 800 °C, the maximum power density of SOFCs ranged between 0.58 and 1.2 W/cm^2^ depending on the electrolyte deposition mode. Scanning electron microscopy was carried out to investigate the surface morphology and structure of the YSZ electrolyte films after thermal annealing. Additionally, an X-ray diffraction analysis of the YSZ electrolyte films was conducted for the synchrotron radiation beam during thermal annealing at different temperatures up to 1300 °C. It was found that certain deposition modes provide the formation of the YSZ electrolyte films with acceptable residual stresses (<1 GPa) at room temperature, including films deposited on large area anodes (100 × 100 mm^2^).

## 1. Introduction

Anode-supported planar solid oxide fuel cells (SOFCs) are one of the most widespread types of fuel cells because they are suitable for operations at lower temperatures (below 800 °C) [1] due to ohmic resistance of thin (10 µm) film electrolyte substantially lower than that of electrolyte-supported cells [2,3].

The manufacture of planar SOFCs is problematic in gaining mechanically stable structures since thin-film electrolyte fractures under moderate loads. Hence, the improvement of mechanical properties of solid oxide fuel cells is very important in their production. Information offered in the literature shows that residual stresses mainly occur in the electrolyte layer. Conventionally, anodes are fabricated from nickel oxide/yttria-stabilized zirconia composite, while yttria-stabilized zirconia (YSZ) is used as an electrolyte. Since SOFC stack components are rigidly joined to ensure a reliable contact between the cells and interconnects, the residual stresses can contribute to component damage, thereby decreasing the operation lifetime and efficiency of SOFCs [4].

Residual stresses can be caused by the difference in the thermal-expansion coefficient of the SOFC layers [5] or an NiO redox behavior in the anode layer [6]. The anode-supported SOFCs (ASCs) are generally fabricated by co-firing the thin-film electrolyte on the anode substrate at 1400–1500 °C [7]. At room temperature, high residual stresses in SOFCs produced by co-firing are mainly caused by the difference in the thermal expansion coefficient [8]. The formation mechanism of residual stresses in co-fired SOFCs has been studied rather well in [7,9,10]. According to these data, the typical values of residual compressive stresses in the electrolyte film for SOFCs vary between 500 and 650 MPa at room temperature. In the electrolyte film, 10 µm thick deposited onto a 0.5 mm thick oxidized anode substrate, the stress level at room temperature is about −600 MPa, while after the temperature raises up to 800 °C, the stresses reduce to −100 MPa [11].

Residual stresses in the magnetron sputtered YSZ film on the SOFC anode remain largely unexplored. This is because information about the reactive magnetron sputtering of YSZ thin films large area anodes (100 × 100 mm^2^ and larger) has appeared only recently [12,13]. In manufacturing non-stacked SOFCs of a 50 × 50 mm^2^ area and smaller, residual stresses and, consequently, cell bending is not so relevant problem. The latter occurs in commercial-grade SOFC manufacture, that is used for stacking and their planarity must meet the high standards.

The magnetron sputtered YSZ electrolyte films were studied in [14,15,16,17,18] and in the state-of-the-art paper [19]. It was shown that the microstructure, phase composition, and other parameters of the YSZ electrolyte films depended on the magnetron sputtering modes, electric supply parameters (RF, DC, pulsed DC), working pressure, substrate temperature, substrate bias voltage, and post-annealing treatment. However, the influence of these parameters on the residual stresses in the YSZ electrolyte films was not investigated.

The residual stress in the magnetron sputtered YSZ electrolyte films can be both intrinsic and thermal. The former is caused by ion, neutral atom fluxes, while the latter results from the difference in the thermal expansion coefficient of the YSZ film and substrate in the case of the higher-temperature film deposition.

Such defects as interstitial atoms, vacancies, dislocations contribute much to the intrinsic stresses in sputter-deposited thin electrolyte films [20]. For example, in TiN films produced by high-power impulse magnetron sputtering, residual compressive stresses are about 10 GPa [21]. According to D’Heurle and Harper [20], one of the main mechanisms of stress formation in thin films deposited by high-energy particles is atomic or ion peening [22]. After a collision with the growing film, some of the adatoms penetrate its surface, in the sites of the crystal lattice. This leads to lattice distortion and excess film compaction. The resulting compressive stresses in the film are proportional to its molar volume and elastic modulus [23].

Residual stresses can be effectively reduced by high-temperature annealing, which is often used in practice [24]. Even though the high-quality YSZ electrolyte films are deposited at room temperature, their successive thermal processing increases their ion conductivity due to the improved crystallinity and density [25,26,27].

In this work, we investigate the variation of residual stresses in 4–6 μm thick-film electrolytes after reactive magnetron sputtering and thermal annealing in air. Reactive magnetron sputtering of thin electrolyte films has been intensively investigated since around 1980. Metal target sputtering in a reactive gas (oxygen) facilitates the formation of oxide films [28]. This method makes it possible to obtain sufficiently dense electrolytes possessing good adhesion. The target surface sputtering in a mixture of Ar and O_2_ gases is the main formation mechanism of sputter-deposited particle fluxes. Depending on the oxygen concentration in the vacuum chamber, the deposition process can be performed in three modes, namely metallic, transition, and oxide [29]. Threshold values of deposition modes determined by the oxygen concentration depend on the sputtered material, vacuum chamber size [30] and pumping speed [31], magnetron configuration (cylindrical or planar) [32], target size [33], power supply, target wear, and other parameters. The metallic mode implies the metal atom sputtering by argon ions at the highest deposition rate. In the transition mode, the target surface is partially covered with an oxide layer, whose thickness is uncontrollable. The instability of the deposition process parameters results in a bad reproducibility if not to adopt such special measures as a closed-loop control system with an optical emission spectrometer [34]. The transition mode provides the film electrolyte sputtering with oxygen deficiency. In the oxide mode, the target surface is almost completely coated with the oxide layer leading to a drastic decrease in the electrolyte deposition rate. The film is deposited with the oxide content approaching stoichiometry. In practice, all three modes are used. Each mode has its own strengths and weaknesses depending on the purpose and requirements for the coating properties. In this work, we compare the YSZ electrolyte films obtained in the transition and oxide modes at different deposition rates and oxygen concentrations.

Half-cell bending after the electrolyte deposition can cause problems during the cathode screen printing or stack assembly. The aim of this work is to analyze the residual stresses in magnetron sputtered thin-film electrolytes, minimize the residual stresses and half-cell bending.

## 2. Materials and Methods

The anode substrates 100 × 20 mm^2^ in size were prepared by laser cutting from 100 × 100 mm^2^ commercial anodes 700 μm thick (Kceracell Co., Chungcheongnam-Do, Korea). The electrolyte deposition process was described in [13]. It was a rather slow process, which provided a precise control for the electrolyte layer thickness and allowed fabricating very dense electrolyte films for SOFCs. The anode substrates were installed on a rotating table, in front of the dual planar magnetrons, as illustrated in Figure 1. Sputtering of 99.5% purity Zr-Y targets (Girmet, Moscow, Russia) 100 × 300 mm^2^ in size was conducted in a mixture and Ar and O_2_ gases. The Zr/Y ratio was 85:15 at.%. The distance between the targets and substrates was ~90 mm. The vacuum chamber was evacuated with a diffusion pump down to a residual pressure of 0.01–0.03 Pa. The substrates were then heated to 450 °C, which was maintained during the YSZ electrolyte film deposition. For better film adhesion, the substrates were treated by Ar ions generated by the ion source with the anode layer. During the YSZ film deposition, the total pressure in the chamber was 0.2 or 0.4 Pa. The stable film deposition was provided by the pulsed power supply APEL-M-12DU-symmetric (Applied Electronics, Tomsk, Russia) operating at 50 kHz frequency and 4 kW constant power. The power supply had a feedback system with an oxygen flow control unit, which allowed the user to keep constant the discharge voltage. The argon gas rate was also constant (40–50 sccm).

Three modes used for the YSZ electrolyte film deposition differ by the oxygen concentration in the vacuum chamber and, consequently, the deposition rate. Figure 2 presents the typical dependence between the rates of the oxide-film deposition and the oxygen gas flow; each mode is indicated by letters *A*, *B*, and *C* [35]. At *A* point of the transition mode observed near the metallic mode, the deposition rate is the highest (35–45 nm/min), whereas the oxygen concentration is the lowest. The latter is manifested in the low electrolyte film transparency in the visible region (see Figure 2b). At *B* point observed at the center of the transition mode, the deposition rate is 16–25 nm/min, and a slight oxygen deficiency, which is proven by the high electrolyte film transparency. At *C* point of the oxide mode, the deposition rate is the lowest (6 or 7 nm/min), its composition being close to the stoichiometry.

After the YSZ electrolyte film deposition, some of the substrates were subjected to 1200 °C annealing in air for 2 h in an oven (Nabertherm, Lilienthal, Germany). The ramp-up rate was controlled at 3 °C/min and then the YSZ anode substrates were free-cooled in the oven. 

The residual stress σ_*f*_ in the YSZ electrolyte films is determined by the substrate bending from Stoney’s formula [36]:(1)$$\sigma_{f}=\frac{E_{s}\cdot t_{s}^{2}}{6\cdot \left(1-\nu_{s}\right)\cdot R_{s}\cdot t_{f}}$$
where *E*_s_ is the elastic modulus of the substrate, viz., 210 GPa [37]; *t*_s_ is the substrate thickness; *ν*_s_ is the Poisson number of the substrate, viz., 0.3 [38]; *R*_s_ is the bend radius of substrate; *t*_f_ is the film thickness.

The bend radius in Stoney’s formula can be calculated from
(2)$$R_{s}=\frac{R_{1}\cdot R_{2}}{R_{1}-R_{2}}$$
where *R*_1_ is the bend radius of the uncoated substrate; *R*_2_ is the bend radius of the coated or annealed substrate.

The bend radius of the YSZ anode substrate is calculated from *R* = (*l*/2)^2^/2*h*, where *l* is the substrate length and *h* is the substrate height. A schematic of the residual bending stress is illustrated in Figure 3.

Malzbender et al. [11] reported that the values of the residual bending stresses in the electrolyte layer coincided with those determined by the X-ray diffraction (XRD) analysis.

High-resolution scanning electron microscopy (SEM) was carried out with a Supra 50VP microscope equipped with the INCA Energy+ microanalysis system (Zeiss, Oberkochen, Germany) to investigate the microstructure of the deposited electrolyte film. The surface morphology was investigated on a Quanta 200 3D dual beam system (FEI Company, Hillsboro, OR, USA).

An MII-4 micro-interferometer (LOMO, Saint Petersburg, Russia) and cross-sectional SEM images were used to measure the thickness of the electrolyte films.

The XRD analysis of the YSZ electrolyte film was conducted for the synchrotron radiation (SR) beam of the VEPP-3 storage ring at the Siberian Synchrotron and Terahertz Radiation Centre of the Budker Institute of Nuclear Physics SB RAS, Novosibirsk, Russia. During this analysis, the YSZ coated substrate was heated from 30 to 1300 °C at a 15 °C/min velocity. In operation, the SR wavelength was 0.172 nm. The 2θ diffraction angle of X-ray radiation was recalculated to the angle at 0.1541 nm wavelength (Cu *K*_α_ radiation) for comparing the obtained results with those measured by a commercial X-ray diffraction apparatus.

The properties of SOFCs with the magnetron sputtered YSZ electrolyte were studied on disk-shaped samples with a diameter of 20 mm cut by laser from 100 × 20 mm^2^ YSZ coated anodes. The La_0.6_Sr_0.4_CoO_3_ cathode (Kceracell Co., Chungcheongnam-Do, Korea) with a 10 × 10 mm^2^ active area was screen printed onto the YSZ electrolyte film and fired in situ during the cell test start-up. Electrochemical investigations were performed at 800 °C, supplying dry hydrogen to the anode and air to the cathode at constant rates of 120 and 350 mL/min, respectively. Ag mesh and Pt wires were used for the current collection from the anode and cathode.

## 3. Results

Table 1 contains deposition process parameters and residual stress values for the YSZ electrolyte film deposition. Films 1–3 are obtained at 0.4 Pa pressure at *A*, *B*, *C* points indicated in Figure 2a. After the YSZ electrolyte film deposition, all half-cells warp toward the electrolyte side, as shown in Figure 4. This proves the presence of residual compressive stresses common to sputter-deposited oxide films [39]. In the electrolyte films fabricated in the transition mode, i.e., with the oxygen deficiency, moderate stresses of 670–690 MPa are detected. In the oxide mode, these stresses are significantly higher (1780 MPa).

As presented in Figure 5, the YSZ electrolyte films obtained at *B* and *C* points (see Figure 2a) do not change and remain transparent after 1200 °C annealing in air. As for point *A*, the film obtained with the large oxygen deficiency becomes opaque after annealing. This proves the light scattering on optical film defects. Since oxygen has a high solubility in zirconium, it can be easily inserted into the oxygen-deficient phase of zirconia.

After 1200 °C annealing in air, the residual stresses in the YSZ film fabricated in the oxide mode (film 1) reduce by 2 times and equal 947 MPa. The residual stresses reduce to 166 MPa in the YSZ film with small oxygen deficiency (film 2). This is probably associated with the diffusion of the interstitial atoms in the lattice sites, which reduces the lattice distortion. Moreover, Quinn et al. [40] indicate other possible reasons for the decrease in the residual stresses after annealing. First, the as-deposited film possesses a mixed crystalline-amorphous structure, which crystallizes after annealing. Second, a change in the crystallographic phase accounts for different crystal phases in the sputter-deposited YSZ film [41]. Third, the sputtering gas is released from the YSZ film.

On the contrary, the residual stresses in the annealed YSZ film fabricated with a large oxygen deficiency (film 3), increase by 10 times, up to 6631 MPa. This is probably because interstitial O_2_ atoms strongly distort the crystal lattice during the annealing process.

The surface morphology and structure of the YSZ films after the different deposition modes are investigated after 1200 °C annealing. SEM images of these films are presented in Figure 6. According to Thornton model [42] describing the film structure depending on the substrate temperature *T* and melting temperature *T*_m_, the films obtained at *T*/*T*_m_ = 0.2–0.3 possess a columnar structure with voids along the grain boundaries inside the columns and between them. This columnar structure is typical of all YSZ electrolyte films deposited by magnetron sputtering [12,17,30]. In our early research [43], we however showed that after high-temperature annealing, the columnar structure merges and becomes totally indistinguishable due to the mass diffusion inside the YSZ films at the sintering stage.

As can be seen in Figure 6, the deposition mode strongly affects the film surface morphology and structure after its annealing. The film obtained in the oxide mode (Figure 6a) has a grained surface comprising 2 µm rounded grains, which, in turn, consist of finer subgrains. The grains constituting the YSZ film are closely adjacent to each other. The cross-section in Figure 6b shows that the film annealing results in surface recrystallization, i.e., the columnar grains join to a denser structural formation. Nevertheless, one can see spherical and elongated closed pores resulting from recrystallization at the columnar grain boundaries.

The surface of the YSZ film deposited at a small oxygen deficiency (Figure 6c) consists of polygonal grains with rounded edges, which are loosely adjacent to each other. The section in Figure 6d shows a lot of closed pores, which are larger than those observed in Figure 2b.

The surface of the YSZ film deposited at a large oxygen deficiency (Figure 6e) is nonuniform and consists of rounded grains. The section in Figure 6f demonstrates two layers of different density. The bottom layer is dense, with closed pores. The structure of the upper layer is highly porous, with open channels and pores. The difference between these layers can probably be explained by the following. During the film deposition, an uncontrolled variation of the magnetron discharge parameters resulted in the shift (along the curve shown in Figure 2a) towards the region with the low oxygen flow rate, i.e., the YSZ film was deposited with the oxygen deficiency larger than it was anticipated. Despite this oxygen deficiency, the film annealing on substrate *3* did not result in its cracking that often occurred after the Zr film oxidation owing to great changes in the film volume [44]. Coddet et al. [18] found that the volume expansion during the zirconium-to-zirconia transition was 1.57. 

To identify the effect of the working pressure in the chamber on residual stresses, the YSZ electrolyte films 4 and 5 were obtained at 0.2 Pa pressure. In the transition mode at a deposition rate of 25 nm/min, the films demonstrated the lowest residual stresses of 455 and 505 MPa, respectively for films 4 and 5. However, similar to film 3, the annealed film 4 manifested 6336 MPa residual stress, resulting in strong bending of the substrate. 

In order to restrain the substrate bending, film 5 is annealed under the dead load. Such an approach is often used in sintering laminated ceramics [45,46]. For this, the coated substrate is placed between two 120 × 120 × 20 mm^3^ flat alumina plates (600 g) during sintering. As can be seen from Table 1, residual stress in film 5 is 1764 MPa, which is 3.5 lower than for film 4 fabricated in the same conditions.

Dead-load annealing was used for film 6 with the YSZ film deposited at a 35 nm/min rate. The residual stress in the deposited film was 930 MPa, while after the dead-load annealing it reduced to 371 MPa.

The SOFC evenness remaining after the YSZ film deposition is the important requirement. It is also important to retain the gas impermeability for the electrolyte layer. Therefore, we compare the volt-ampere characteristics of SOFCs fabricated from the YSZ electrolyte films 1–6 subjected to annealing and given in Table 1 and Figure 7. The results obtained at 800 °C are summarized in Table 2; hydrogen and air are fuel and oxidizing medium, respectively.

The open-circuit voltage (OCV) in SOFC with the gas-impermeable electrolyte usually ranges from 1.08 to 1.1 V. Lower OCV values indicate cracks or through pores in the YSZ electrolyte film, which lead to a direct contact between the fuel and oxidizer. According to Table 2 and Figure 7, only SOFC 2 has a low OCV of 816 mV indicating insufficient electrolyte density. This correlates with the YSZ film surface in Figure 6c, where one can see loosely adjacent grains. SOFC 2 demonstrates the lowest power density of 462 mW/cm^2^ for the same reason. The open-circuit voltage of other fuel cells approaches its theoretical value and can be increased after deposition of the gadolinia-doped ceria barrier layer onto the YSZ electrolyte film.

SOFC 3 has the highest properties among SOFCs 1–3, the YSZ electrolyte for which is obtained with different oxygen nonstoichiometry. Despite the two-layer structure of the electrolyte with different densities of its layers, SOFC 3 has the highest OCV and power density values of 1083 mV and 896 mW/cm^2^, respectively. This deposition mode is more interesting from the practical viewpoint since it provides the highest deposition rate. The dead-load annealing (SOFCs 5 and 6) allows increasing the cell power up to 1000 mW/cm^2^ and higher, the gas impermeability of the YSZ electrolyte film remains. 

In Figure 8a,b, one can see the surface morphology of the YSZ film for SOFC 5 after 1200 °C dead-load annealing. The film surface is dense and after annealing, it is not cracked. This surface consists of submicron triangular grains with faceted sharp edges. The cross-sectional SEM image in Figure 8c shows the dense and uniform structure.

The XRD patterns of deposited YSZ electrolyte films during annealing were recorded on the SR source. The high intensity and collimation of the synchrotron radiation allowed for high-resolution studies. 

Figure 9 shows in situ XRD patterns for film 2 obtained during annealing in air and cooling to room temperature. The as-deposited YSZ electrolyte film displays the typical diffraction pattern of cubic Y_0.16_Zr_0.84_O_1.92_ (ICDD PDF database file number 04-001-9395). The diffraction peaks of 50.2, 59.6, and 62.6 degrees 2θ unambiguously assign the cubic phase corresponding to (220), (311), and (222) planes, respectively. The XRD pattern of tetragonal zirconia demonstrates peak splitting at around 60 degrees 2θ [47]. As compared to the stress-free film, the shift in the peak position indicates that this film is under residual compressive stress. During heating up to 1300 °C in air, the YSZ diffraction peaks gradually shift to the left due to the thermal expansion of the crystal lattice. In these conditions, the decrease in the peak intensity is explained by the drift of the SR beam during the process of measurement. After heating up to 1300 °C, the monochromator is corrected, and the peak intensity increases up to its initial value.

In the film cooling, the diffraction peaks move smoothly back in their initial position. The peak shift in X-ray diffraction of the YSZ film (film 2) before (black) and after (red) 1300 °C annealing in air is presented in Figure 10. After annealing and cooling to room temperature, the peaks slightly shift toward larger angles due to a decrease in the lattice parameter caused by annealing. This indicates that the residual compressive stress amplitude reduces in the film, which is held in agreement with the experimental data summarized in Table 1. After annealing, the peak width decreases indicating the growth in the coherent scattering region in the film, namely ~55–60 nm. After the thermal annealing, the (220) peak intensity increases and the (311) peak intensity lowers. The peak intensity ratio (220)/(311) after 1300 °C annealing grows from 0.92 to 1.02 and is about 1.62 for the unstressed state according to 04-001-9395 ICDD PDF database file number.

To test practical implications, the YSZ electrolyte film 5 µm thick was deposited onto 100 × 100 mm^2^ 700 μm thick anode substrates in the transition mode, at a 34 nm/min deposition rate. The obtained sample was placed between two flat alumina plates 120 × 120 × 20 mm^3^ in size and 600 g weight and then annealed at 1200 °C in air under the additional load of 2000 g. In Figure 11b, one can see that the fabricated anode-supported half-cell remains flat; and in Figure 11a, it is strongly deformed after the formation of 5 µm thick YSZ electrolyte film in the oxide mode followed by the stress-free annealing.

## 4. Conclusions

In this paper, we showed that during reactive magnetron sputtering, the YSZ film deposition rate varied from 6 to 45 nm/min depending on the oxygen concentration in the vacuum chamber. At the same time, the film structure ranged from dense to very porous. Based on the results, it can be concluded that the residual stresses in the magnetron sputtered YSZ electrolyte films were successfully controlled by the deposition mode and annealing. At the same time, the annealing treatment provided the YSZ films with a dense structure and improved their stoichiometry. The obtained results can offer technical feasibility for residual stress control and compensation in the YSZ films. Residual compressive stresses were found in as-deposited YSZ electrolyte films. The stresses depended on the oxygen concentration in the films such that the lowest residual stresses were observed in substoichiometric films, while the highest were in stoichiometric films. The thermal annealing of the YSZ electrolyte films with small or without oxygen deficiency at 1200 °C in air reduced the residual stresses by 2 or 3 times. On the contrary, the YSZ film annealing with high oxygen deficiency resulted in multiple growths in the residual stresses. It was shown that this problem could be eliminated by the dead-load annealing. Volt-ampere characteristics of SOFCs with YSZ electrolyte films depended on their deposition and annealing modes. These parameters allowed the maximum power density of SOFCs to range within 0.58–1.2 W/cm^2^ at 800 °C. Thus, from the practical point of view, it was more efficient to deposit the YSZ films in the transition mode, when the deposition rate is much higher than that in the oxide mode. The resulting YSZ films acquired a cubic crystal lattice, had low residual stresses, and had a gas-impermeable structure.

## Figures and Tables

**Figure 1 membranes-12-00346-f001:**
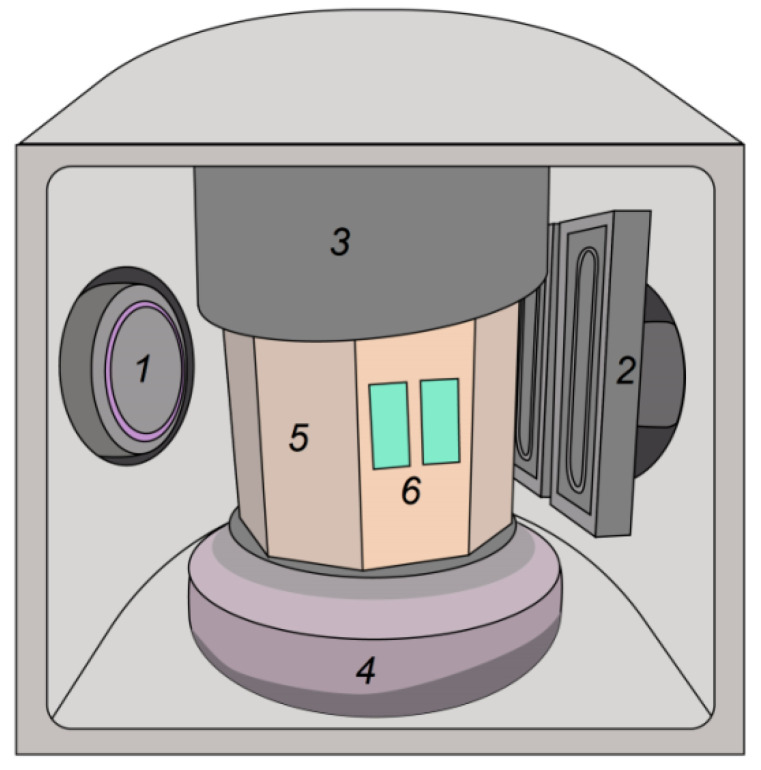
Film deposition facility: 1—anode-layer ion source, 2—dual magnetrons, 3—shield, 4—rotating table, 5—drum with heaters, 6—anode substrates.

**Figure 2 membranes-12-00346-f002:**
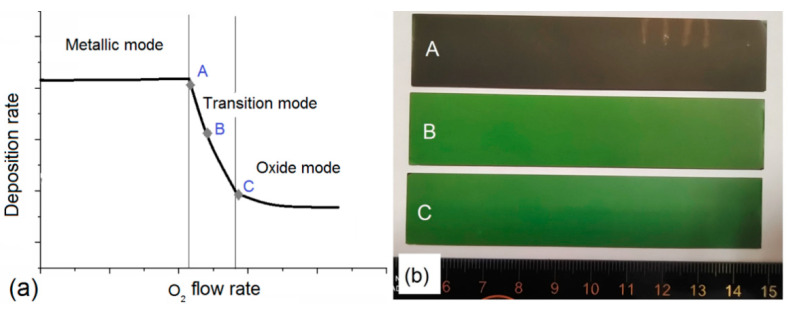
Dependence between YSZ electrolyte film deposition and oxygen flow rates (**a**); a photograph of electrolyte films deposited at *A*, *B*, *C* points (**b**).

**Figure 3 membranes-12-00346-f003:**
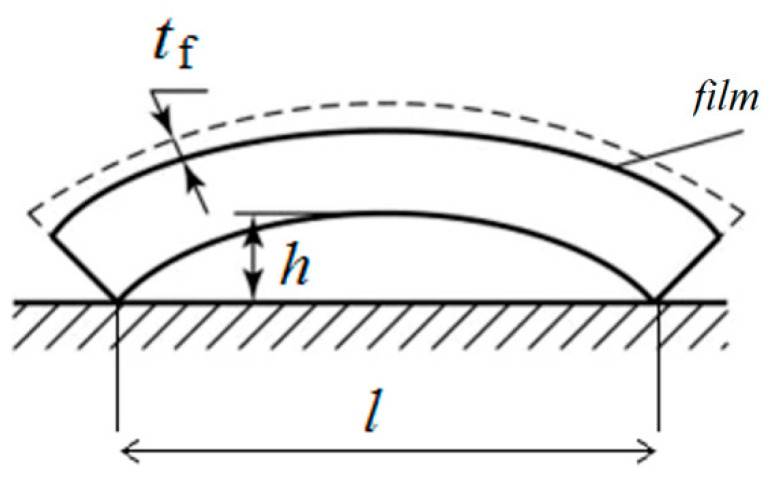
Schematic of residual bending stress.

**Figure 4 membranes-12-00346-f004:**
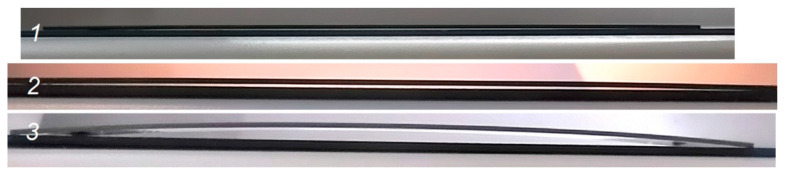
Side view of substrates: 1—initial anode substrate; 2—after YSZ electrolyte film deposition, 3—after 1200 °C annealing; YSZ electrolyte is deposited on top.

**Figure 5 membranes-12-00346-f005:**
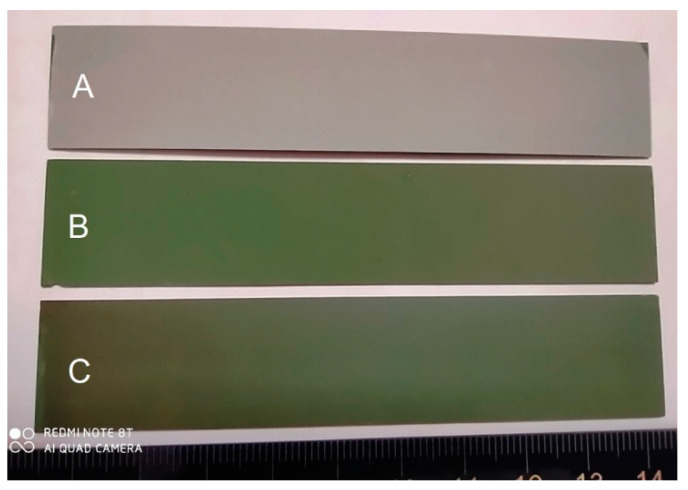
Photograph of YSZ electrolyte films (from Table 1) annealed at 1200 °C: *A*—film *3*, *B*—film *2*, *C*—film 1.

**Figure 6 membranes-12-00346-f006:**
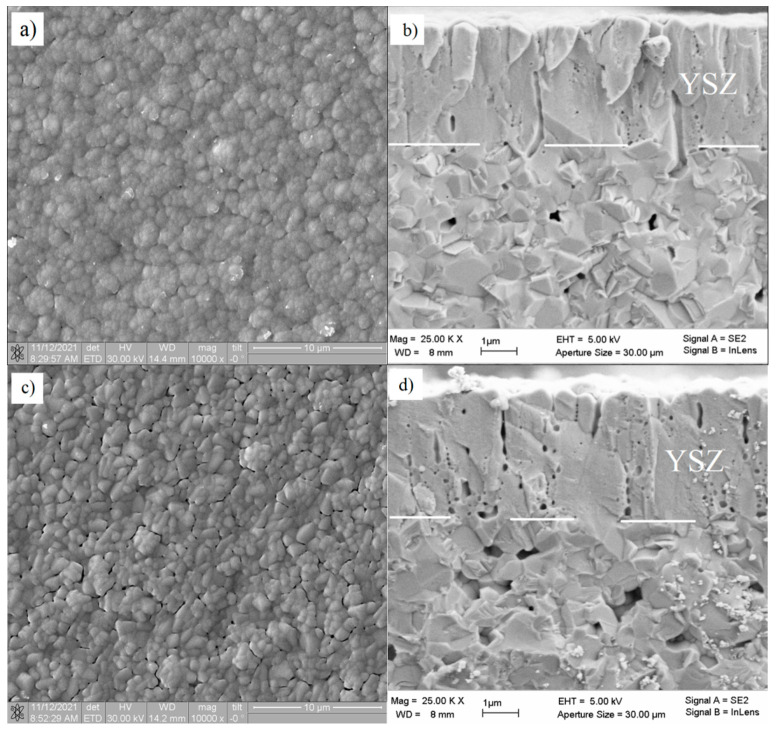
SEM images YSZ film surface and cross-section after 1200 °C annealing: (**a**,**b**)—film 1, (**c**,**d**)—film 2, (**e**,**f**)—film 3.

**Figure 7 membranes-12-00346-f007:**
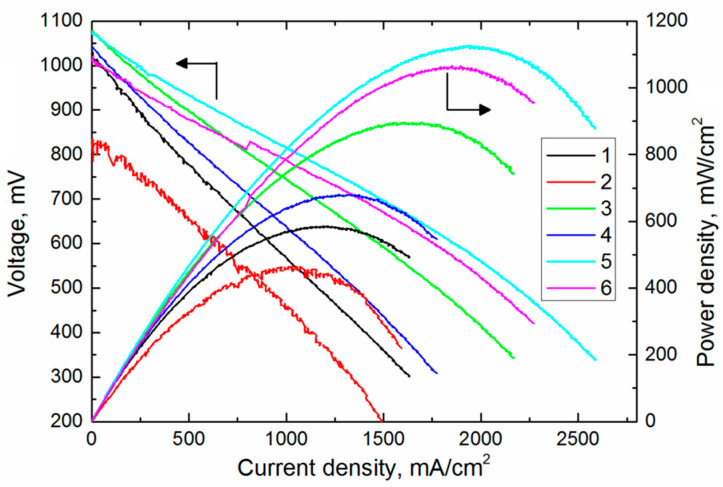
Volt-ampere characteristics of SOFCs fabricated from YSZ electrolyte films 1–6 at 800 °C.

**Figure 8 membranes-12-00346-f008:**
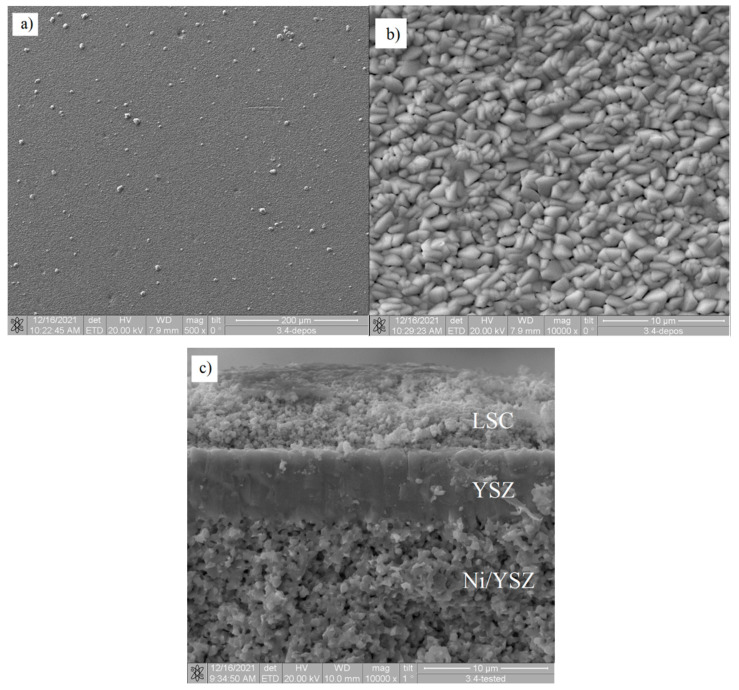
SEM images of YSZ film 5 after 1200 °C dead-load annealing: (**a**,**b**)—surface, (**c**)—cross-section.

**Figure 9 membranes-12-00346-f009:**
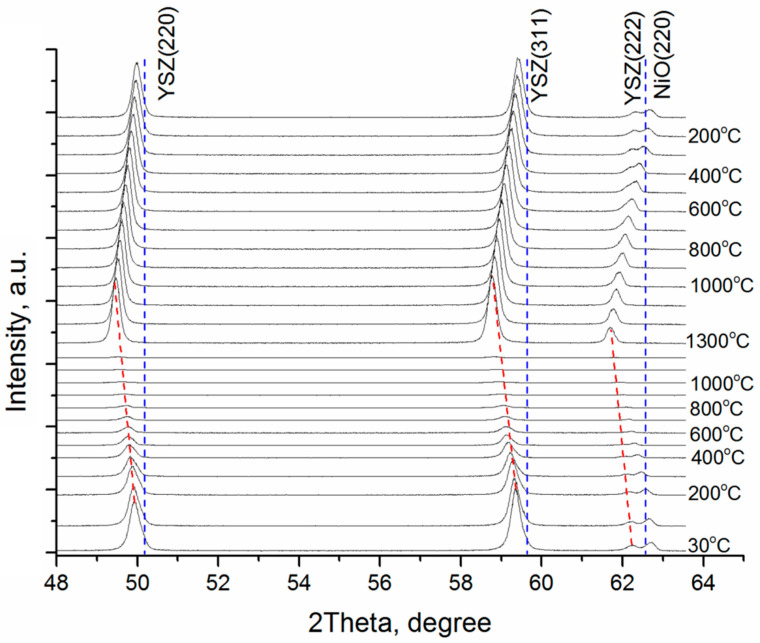
XRD patterns collected on the synchrotron beam line from YSZ film 2 after 1300 °C annealing in air and cooling to room temperature (8YSZ, PDF Card 04-001-9395 references are reported for comparison).

**Figure 10 membranes-12-00346-f010:**
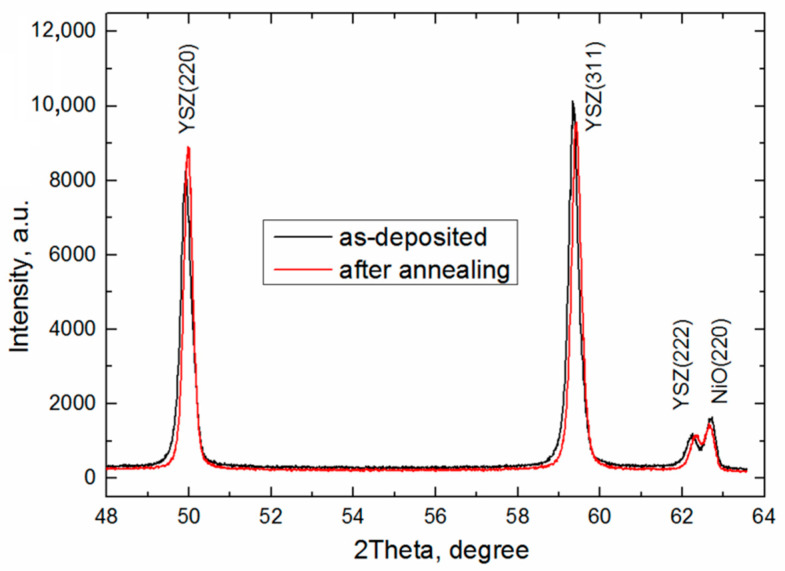
Peak shift in X-ray diffraction of YSZ film (SOFC 2) before (black) and after (red) 1300 °C annealing in air.

**Figure 11 membranes-12-00346-f011:**
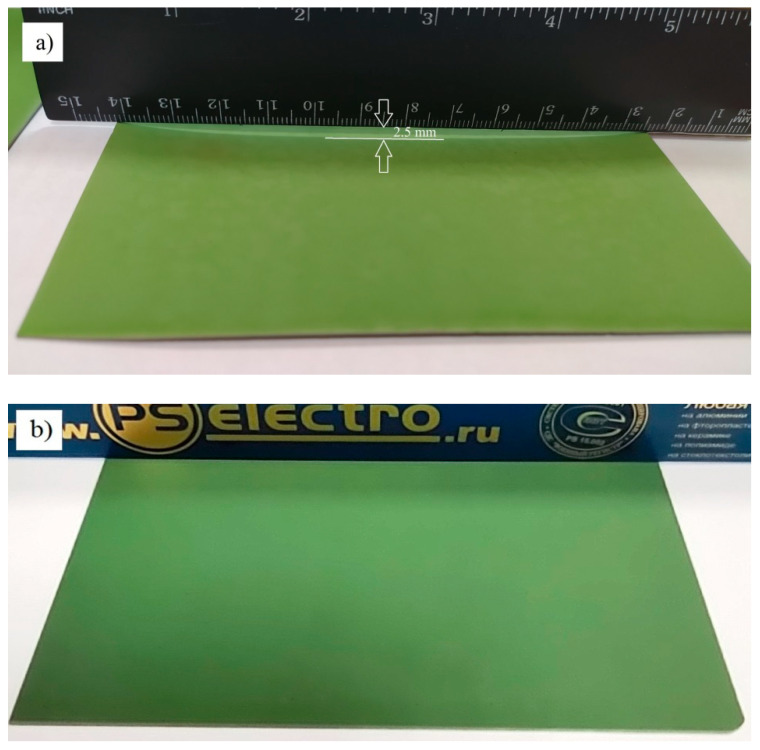
Photographs of ASCs with YSZ electrolyte film deposition: (**a**) in oxide mode followed by stress-free annealing; electrolyte on ASC underside, (**b**) in transition mode and annealed under load; electrolyte on ASC top.

**Table 1 membranes-12-00346-t001:** Deposition process parameters and residual stresses for YSZ electrolyte films.

YSZ Electrolyte Films	Deposition Modes	Pressure, Pa	Deposition Rate, nm/min	Film Thickness, µm	Residual Stress, MPa	
					After Deposition	After 1200 °C Annealing
1	Oxide	0.4	6.1	4	1780	947
2	Transition	0.4	16	4	690	166
3	Transition	0.4	45	5	670	6631
4	Transition	0.2	25	6.1	455	6336
5	Transition	0.2	25	6.1	505	1764 ^1^
6	Transition	0.2	35	4.7	930	371 ^1^

^1^ A loading plate of 600 g is placed on each half-cell top during annealing.

**Table 2 membranes-12-00346-t002:** Parameters of SOFCs with thin film YSZ electrolyte at 800 °C.

SOFCs	Open Circuit Voltage, mV	Maximum Power Density, mW/cm^2^
1	1043	586
2	816	462
3	1083	896
4	1051	678
5	1082	1225
6	1057	1063

## Data Availability

The data presented in this study are available on request from the corresponding author.

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
