# Peer review of "Influence of Deposition Modes and Thermal Annealing on Residual Stresses in Magnetron-Sputtered YSZ Membranes"

_membranes, 2022, doi:10.3390/membranes12030346_

Round 1

Reviewer 1 Report

I think the authors did a wonderful job on preparing YSZ thin films with the magnetron sputtering method. This work will contribute to the community of SOFC significantly. I have 2 comments to make.

  1. In Line 88-94, the authors listed three modes of sputtering. What are the threshold values of these modes in terms of oxygen partial pressure? The authors may have to clarify this based on existing literature.
  2. Are authors using a Zr-Y alloy target during the sputtering? Or pure Zr and pure Y targets.

Author Response

Responses to Reviewer #1’s comments:

  1. COMMENT: In Line 88-94, the authors listed three modes of sputtering. What are the threshold values of these modes in terms of oxygen partial pressure? The authors may have to clarify this based on existing literature.

RESPONSE: In this paper we intentionally did not give the threshold values of three modes of sputtering in terms of oxygen partial pressure, because they are specific to each vacuum system and depend on many factors: the sputtered material, the size of the vacuum chamber [1] and its pumping speed [2], the magnetron design (cylindrical or planar) [3], the target size [4], the type of power supply, the degree of target wear, etc. The information has been added to the paper.

  1. Spencer, A. G., & Howson, R. P. System Design For High Rate Deposition Of Indium Oxide Solar Coatings . Optical Materials Technology for Energy Efficiency and Solar Energy Conversion VII. (1989). Proceedings Volume 1016 doi:10.1117/12.949931
  2. Okamoto, A., & Serikawa, T. (1986). Reactive sputtering characteristics of silicon in an Ar-N2 mixture. Thin Solid Films, 137(1), 143–151. doi:10.1016/0040-6090(86)90202-6
  3. Kelly, P. J., West, G., Kok, Y. N., Bradley, J. W., Swindells, I., & Clarke, G. C. B. (2007). A comparison of the characteristics of planar and cylindrical magnetrons operating in pulsed DC and AC modes. Surface and Coatings Technology, 202(4-7), 952–956. doi:10.1016/j.surfcoat.2007.04.130
  4. Nyberg, T., Berg, S., Helmersson, U., & Hartig, K. (2005). Eliminating the hysteresis effect for reactive sputtering processes. Applied Physics Letters, 86(16), 164106. doi:10.1063/1.190633

COMMENT: 2. Are authors using a Zr-Y alloy target during the sputtering? Or pure Zr and pure Y targets.

RESPONSE: We used 99.5% purity Zr-Y targets with the Zr/Y ratio of 85:15 at.%. This information is in the paper in section 2. Materials and Methods.

Reviewer 2 Report

In the current manuscript YSZ membranes are deposited over the surface of SOFC anode using Magnetron sputtering. The results/discussion are well described and supported by the characterizations necessary for SOFC application. However minor revisions are required before acceptance; 1. The authors must mention peak electrochemical performance in the abstract. 2. Addition of flow chart of experimentation will make this article more interesting to readers. 3. The authors should provide electrochemical performance plot (OCV and Power Density w.r.t current density) 4. The format of citation numbering should be the same. 5. The conclusion is not providing all observations therefore authors should rewrite it. 6. There should be some citations from 2019-2021, especially in the introduction. The following recent articles will be good addition in the Introduction section. https://doi.org/10.1016/j.ceramint.2020.09.140

Author Response

Responses to Reviewer #2’s comments:

In the current manuscript YSZ membranes are deposited over the surface of SOFC anode using Magnetron sputtering. The results/discussion are well described and supported by the characterizations necessary for SOFC application. However minor revisions are required before acceptance;

  1. COMMENT: The authors must mention peak electrochemical performance in the abstract.

RESPONSE: We added peak electrochemical performances in the abstract.

  1. COMMENT: Addition of flow chart of experimentation will make this article more interesting to readers.

RESPONSE: Flow charts usually explain a complicated process by organizing the information visually and breaking up each step or element of the process. Our work does not use multistage processes, so we consider the use of flowcharts unnecessary.

  1. COMMENT: The authors should provide electrochemical performance plot (OCV and Power Density w.r.t current density)

RESPONSE: We added electrochemical performance plots (Fig. 7).

  1. COMMENT: The format of citation numbering should be the same.

RESPONSE: We tried to use the format of citation numbering according to the requirements of the journal.

  1. COMMENT: The conclusion is not providing all observations therefore authors should rewrite it.

RESPONSE: In conclusion, some additions have been made.

  1. COMMENT: There should be some citations from 2019-2021, especially in the introduction. The following recent articles will be good addition in the Introduction section. https://doi.org/10.1016/j.ceramint.2020.09.140

RESPONSE: We added some citations the Introduction section.